# Estimation of Winter Wheat Canopy Chlorophyll Content Based on Canopy Spectral Transformation and Machine Learning Method

Xiaokai Chen, Fenling Li, Botai Shi, Kai Fan [ORCID], Zhenfa Li and Qingrui Chang *

College of Natural Resources and Environment, Northwest A&F University, Yangling 712100, China
* Correspondence: changqr@nwsuaf.edu.cn; Tel.: +86-135-7183-5969

**Abstract:** Canopy chlorophyll content (CCC) is closely related to crop nitrogen status, crop growth and productivity, detection of diseases and pests, and final yield. Thus, accurate monitoring of chlorophyll content in crops is of great significance for decision support in precision agriculture. In this study, winter wheat in the Guanzhong Plain area of the Shaanxi Province, China, was selected as the research subject to explore the feasibility of canopy spectral transformation (CST) combined with a machine learning method to estimate CCC. A hyperspectral canopy ground dataset in situ was measured to construct CCC prediction models for winter wheat over three growth seasons from 2014 to 2017. Sensitive-band reflectance (SR) and narrow-band spectral index (NSI) were established based on the original spectrum (OS) and CSTs, including the first derivative spectrum (FDS) and continuum removal spectrum (CRS). Winter wheat CCC estimation models were constructed using univariate regression, partial least squares (PLS) regression, and random forest (RF) regression based on SR and NSI. The results demonstrated the reliability of CST combined with the machine learning method to estimate winter wheat CCC. First, compared with OS-SR (683 nm), FDS-SR (630 nm) and CRS-SR (699 nm) had a larger correlation coefficient between canopy reflectance and CCC; secondly, among the parametric regression methods, the univariate regression method with CRS-NDSI as the independent variable achieved satisfactory results in estimating the CCC of winter wheat; thirdly, as a machine learning regression method, RF regression combined with multiple independent variables had the best winter wheat CCC estimation accuracy (the determination coefficient of the validation set ($R_v^2$) was 0.88, the RMSE of the validation set ($RMSE_v$) was 3.35 and relative prediction deviation (RPD) was 2.88). Thus, this modeling method could be used as a basic method to predict the CCC of winter wheat in the Guanzhong Plain area.

**Keywords:** precision agriculture; winter wheat; canopy chlorophyll content; canopy spectral transformation; narrow-band spectral index; hyperspectral remote sensing

## 1. Introduction

Chlorophyll, carotenoid, and anthocyanin are the three most important pigments in plants [1]. Chlorophyll is the most abundant and important component for green plants to absorb photosynthetically active radiation [2–4]. Moreover, chlorophyll content is closely related to the crop nitrogen status, crop growth and productivity, detection of diseases and pests, and final yield [5–8]. Thus, accurate monitoring of chlorophyll content in crops is of great significance for decision support in precision agriculture.

The traditional methods for determining chlorophyll content include spectrophotometry, high-performance liquid chromatography, and atomic absorption by field sampling and laboratory analysis, which are laborious, destructive to leaves, and time-consuming [9–12]. Several rapid and accurate monitoring studies of plant canopy chlorophyll content (CCC) were carried out using hyperspectral remote sensing technology. However, when using hyperspectral remote sensing data, the redundancy of hyperspectral data and the autocorrelation between spectra will increase significantly, leading to certain data disasters [13].

Additionally, it brings new difficulties to data transmission and processing. In response to the emergence of these problems, previous researchers have established some traditional vegetation indices (VIs) and developed a series of narrow-band spectral indices (NSI) to monitor physiological and biochemical parameters. Dash et al. [14] specifically targeted the medium resolution imaging spectrometer (MERIS) sensor mounted on the Envisat satellite launched by ESA in 2004, and proposed the MERIS Terrestrial Chlorophyll Index (MTCI) for monitoring the chlorophyll content of crops; Broge et al. [15] analyzed the winter wheat canopy hyperspectral data based on different nitrogen levels and suggested the ratio vegetation index (RVI) could effectively predict the CCC. However, these VIS were determined according to specific spectral reflectance. Due to the influence of many factors such as year, crop variety, climate, etc., VIs cannot be directly promoted; they must be calibrated and optimized for specific data sets. Many researchers have carried out a lot of research on the optimization of VIs [16–18]. Zhang et al. [19] used random band combinations to optimize the published vegetation index to estimate the chlorophyll content (CCC) of the winter wheat canopy through the original spectral (OS) and first-order differential (FD) processing. The research results showed the exponential expression formula $R_1/(R_2 \times R_3)$ was an effective choice for monitoring crop agronomic parameters. A large number of studies on maize, wheat, rice, cotton, and barley showed NSI could improve the explanatory power of canopy N concentration by 19–43% compared with the published VIs [20–22].

However, previous studies relied on the original spectrum (OS) to extract NSIs. Because the detection of winter wheat canopy reflectance and the selection of sensitive bands may be affected by external factors such as atmospheric conditions and soil background, the applicability and reliability of the final monitoring model were compromised. Some researchers attempted to weaken the influence of external factors by spectral transformation based on OS [19,23–25]. Canopy spectral transformation is mainly a mathematical transformation of OS to expand the difference between spectral curves under different conditions, so as to improve the sensitivity of sensitive bands. The studies by Li et al. [26] showed that the log-transformed spectrum (LOGS) and continuum removal spectrum (CRS) could improve the remote sensing estimation effects of nitrogen to different degrees compared to the OS. Ta et al. [27] made three transformations on the OS of apple tree leaves, including the reciprocal transformed spectrum (RS), first-order differential spectrum (FODS), and CRS, combining transformations with machine learning methods to estimate the leaf chlorophyll content. The results showed the random forest regression model had the best prediction accuracy for the first growth period ($R^2$ = 0.96, RMSE = 0.95). Although NSI and spectral transformation technology have been successfully applied in previous studies and achieved satisfactory results. However, due to the influence of different crop varieties, growth environment, and growth period, NSI cannot be directly applied. In other words, NSI can only be used after being localized. Thus, the contribution of spectral conversion technology and NSI to the accuracy of the CCC estimation of winter wheat in Guanzhong Plain of Shaanxi Province is worth exploring.

Recently, it has been very popular to combine machine learning with remote sensing data to estimate crop parameters [28–30]. Yuan et al. [31] used the partial least squares regression, an artificial neural network, random forest regression, and support vector machine regression to monitor the leaf area index of the soybean. The results indicated that the random forest regression could display a more accurate result. The work of Wang et al. [32] revealed random forest regression of the whole growth season based on the first derivative spectrum providing reasonable accuracy for each growth stage.

In this study, we focused on the relationship between the winter wheat CCC and spectral reflectance, investigating the possibility of estimating CCC by combining CST with the machine learning method. The main objectives were to: (1) analyze the correlation between CCC and sensitive-band reflectance (SR) and NSI under different spectral transformations; (2) construct CCC estimation models with SRs and NSIs by univariate regression, partial least squares (PLS) regression, and random forest (RF) regression based on different

spectral transformations; and (3) compare the models constructed under objective (2) to find an effective model for estimating the winter wheat CCC during the growing season.

## 2. Materials and Methods

### 2.1. Experimental Setup

The winter wheat study was conducted in Qian County (108°07′ E, 34°38′ N; average altitude: 830 m), and experiment station No.1 was located at the Northwest A&F University (108°03′ E, 34°17′ N; average altitude: 454 m) in Yangling, Shaanxi Province, China from 2014 to 2017 (Figure 1). The whole experiment area belongs to the dry farming area in the southern part of the Loess Plateau, and the climate type belongs to the warm temperate zone semi-humid climate. The annual average rainfall is 630 mm, and the precipitation is highly seasonal, mostly in July, August, and September. The soil types in both stations were loam. Winter wheat in the Guanzhong Plain area was sown in mid-October and harvested the following year in mid-June. All cultivars were Xiaoyan 22 [33], a common local variety. Each treatment was repeated twice, and nitrogen fertilizer, phosphate fertilizer, and potassium fertilizer were applied in each treatment as urea, superphosphate, and potassium chloride, respectively. All fertilizers were applied as basic fertilizer before sowing, and no additional fertilizer was applied during the growing season. In addition, four, three, and ten field trials were set up during 2014–2015, 2015–2016, and 2016–2017, respectively, in Qian County. The tillage system at both sites were monoculture. The management method was the same as that for local conventional winter wheat. Table 1 summarizes the experimental design.

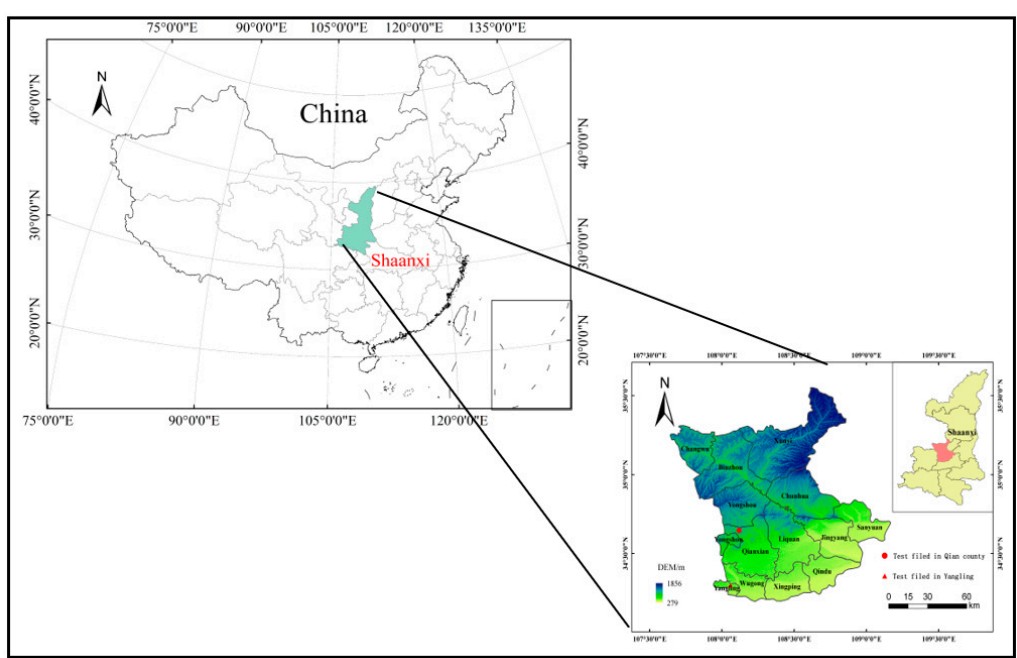

**Figure 1.** Geographical location of the study area.

### 2.2. Canopy Spectral Reflectance Measurement

The canopy spectral reflectance was measured during the main growing season using an SVC HR-1024I portable passive spectrometer developed by the Spectra Vista Company (Poughkeepsie, NY, USA). The portable spectrometer can detect canopy spectral reflectance in the wavelength range of 350–2500 nm. The field of view was 25° and the height to the canopy was 1.3 m. The white calibration panel was performed before and after the measurement [34], and the sensor was pointed vertically downward during the observation. All measurements of canopy spectral reflectance were performed during sunny, windless, and cloudless weather between 10:30 and 14:00 to reduce the error caused by changing illumination conditions. Two sampling points in the diagonal direction of each plot were

selected to take canopy spectral reflectance measurements. The mean value of 10 repetitions was regarded as the final canopy spectral reflectance of each sampling point. The canopy reflectance of each plot was the average of the two sampling points. Real-time kinematic (RTK) was used to mark the coordinates of the sampling sites.

**Table 1.** Summary of the conditions utilized in the winter wheat growth experiments.

| Site | Plot Area (m$^2$) | Sowing Time | Nitrogen Rate (kg·ha$^{-1}$) | Phosphorus (P$_2$O$_5$) Rate (kg·ha$^{-1}$) | Potassium (K$_2$O) Rate (kg·ha$^{-1}$) | Sampling Date |
|---|---|---|---|---|---|---|
| Qian | 36 | 1 October 2014 | 0, 37.5, 75, 112.5, 150, 187.5 | 0, 22.5, 45, 67.5, 90, 112.5 | 0, 15, 30, 45, 60, 75 | 28 March (GS3), 12 April (GS5), 26 April (GS6), 9 May (GS7), 26 May (GS8). (2015) |
| No.1 | 12 | 2 October 2014 | 0, 30, 60, 90, 120, 150 | 0, 15, 30, 45, 60, 75 | | 14 March (GS2), 29 March (GS3), 13 April (GS5), 11 May (GS7), 26 May (GS8). (2015) |
| Qian | 36 | 4 October 2015 | 0, 37.5, 75, 112.5, 150, 187.5 | 0, 22.5, 45, 67.5, 90, 112.5 | 0, 15, 30, 45, 60, 75 | 18 April (GS6), 9 May (GS6), 6 June (GS9). (2016) |
| No.1 | 12 | 5 October 2015 | 0, 30, 60, 90, 120, 150 | 0, 15, 30, 45, 60, 75 | | 14 March (GS2), 29 March (GS3), 13 April (GS5), 11 May (GS7), 26 May (GS8). (2016) |
| Qian | 90 | 1 October 2016 | 0, 30, 60, 90, 120, 150 | 0, 22.5, 45, 67.5, 90, 112.5 | 0, 22.5, 45, 67.5, 90, 112.5 | 26 March (GS3), 14 April (GS5), 28 April (GS6), 17 May (GS7), 26 May (GS8). (2017) |
| No.1 | 12 | 1 October 2016 | 0, 30, 60, 90, 120, 150 | 0, 15, 30, 45, 60, 75 | | 25 March (GS3), 17 April (GS5), 20 May (GS8). (2017) |

Note: Qian and No.1 represent Qian County and experiment No.1 station of Northwest A&F University, respectively. GS2: tillering, GS3: booting, GS5: heading, GS6: flowering, GS7: filling, GS8: ripening, GS9: senescence.

*2.3. CCC Measurement*

After collecting the canopy spectral reflectance, the CCC was measured. Ten wheat plants with the same growth potential were selected near the spectral sampling point, and the relative chlorophyll content of canopy leaves was measured using SPAD-502, which was developed by the Agriculture and Horticulture Bureau, Ministry of Agriculture, Forestry and Fisheries, Japan. Six measurements from the petiole to the tip were taken and averaged as the SPAD values of each canopy leaf. The average SPAD values of the 10 leaves were used as the CCC of the sampling points. The CCC of each plot was determined using the average of the two sampling points.

*2.4. Calibration and Validation*

The CCC was separated into calibration and validation sets in ascending order in a 3:1 ratio to ensure that the range of CCC was balanced [35]. Consequently, 662 and 165 calibration and validation samples were collected, respectively. Table 2 lists the CCC statistical characteristics of the samples. The calibration and validation sets were used to build the CCC estimate models and assess the accuracy of these models. Both the calibration and validation sets obeyed anormal distribution. Across the total growth stages, the CCC varied from 5.78 to 56.43, with a mean value of 45.88. The coefficient of variation (CV) was 21.19%, indicating a moderate degree of dispersion [36].

*2.5. Methods*

2.5.1. Canopy Spectral Transformation (CST)

The reflectance curves of the visible (VIs) and near-infrared (NIR) bands were closely correlated with the CCC and canopy structure of crops [37]. Thus, the canopy spectral

reflectance at 350–1350 nm was selected to estimate the CCC in this study. The Savitzky–Golay smoothing filtering process with a second-order polynomial and nine smoothing points curves were applied to all the canopy reflectance to filter the noise, and the canopy reflectance was resampled to a spectral interval of 1 nm to obtain the OS [38,39]. Finally, the first derivative spectrum (FDS) [40] and continuum removal spectrum (CRS) [41] were determined based on the OS.

**Table 2.** Statistics of CCC characteristics of winter wheat in the whole growth period.

| Data Sets | Number of Samples | Maximum Values | Minimum Values | Mean | Median | SD | CV (%) |
|---|---|---|---|---|---|---|---|
| Calibration set | 662 | 56.43 | 5.78 | 45.88 | 49.41 | 9.73 | 21.21 |
| Validation set | 165 | 55.63 | 7.98 | 45.88 | 49.38 | 9.66 | 21.05 |
| Whole | 827 | 56.43 | 5.78 | 45.88 | 49.40 | 9.72 | 21.19 |

Derivative transformation is the most commonly used transformation form in crop hyperspectral research. Derivative transformation of OS can not only weaken or eliminate the impact of environmental background and atmospheric effects, but also improve the contrast of absorption characteristics of crop biochemical components to varying degrees. The FDS can eliminate the influence noise in the crop canopy spectrum and highlight the characteristic position in the canopy spectrum. The difference method was used to calculate the FDS of the original spectrum, and (1) was the specific calculation formula in this paper.

$$R'_i = \frac{R_{i+1} - R_{i-1}}{2} \tag{1}$$

where $i$ refers to the wavelength of band $i$, $R_i$ refers to the original spectral reflectance corresponding to wavelength $i$, and $R'_i$ refers to the reflectance of FDS corresponding to wavelength $i$.

Kokaly and Clark first proposed continuum removal, also known as envelope removal. The continuum removal spectrum (CRS) was defined as the ratio of the original spectral reflectance to the continuum reflectance of the corresponding band [42]. The CRS can effectively remove spectral information noise and improve the responsiveness of crop nitrogen and chlorophyll [43]. Therefore, the CRS was used as a spectral transformation to perform relevant operations on winter wheat OS in this study.

2.5.2. Canopy Hyperspectral Narrow-Band Spectral Index (NSI)

Hyperspectral remote sensing data from an SVC HR-1024I spectrometer contains a large amount of spectral information. In this study, the simplest and most common vegetation indices at 350–1350 nm were selected; the difference spectral index (DSI), ratio spectral index (RSI), normalized difference spectral index (NDSI), and soil-adjusted spectral index (SASI) were calculated between any two bands of the OS, FDS, and CRS [44]. The calculations are listed in Table 3. The software Matlab2017a, a commercial mathematics software produced by MathWorks (Natick, MA, USA), was used to accomplish the extraction of narrow-band spectral index (NSI).

**Table 3.** NSI and computational formulas.

| NSI | Computational Formulas |
|---|---|
| RSI | $R_i / R_j$ |
| DSI | $R_i - R_j$ |
| NDSI | $\left(R_i - R_j\right) / \left(R_i + R_j\right)$ |
| SASI | $1.5 \times \left(R_i - R_j\right) / \left(0.5 + R_i + R_j\right)$ |

where $i$ and $j$ are the hyperspectral wavelength/nm, and $R_i$ and $R_i$ represent the hyperspectral reflectance corresponding to the wavelengths of $i$ and $j$.

### 2.5.3. Modeling Methods

In this study, univariate regression, partial least squares (PLS) regression, and random forest (RF) regression were applied to estimate the winter wheat CCC.

Univariate regression means a method in which only one independent variable correlated with one dependent variable [45]. In this paper, sensitive-band reflectance (SR) and narrow-band spectral index (NSI) were used as independent variables to build the CCC univariate regression model, respectively, and the exponential, linear, logarithmic, polynomial, and power functions were involved.

Partial least squares (PLS) regression is a regression approach [46] for assessing multivariate statistical data. It maximizes the covariance between the latent variable (LV) and response variable by reducing the input data to certain independent latent variables [47]. The number of LV was determined using the standard error of the leave-one-out cross-validation [48,49]. PLS accounts for strongly linearly dependent independent variables when the number of samples is fewer than the number of variables [50]. The software Matlab2017a, a commercial mathematics software produced by MathWorks (Natick, MA, USA), was used to accomplish the modeling and parameter optimization.

Random forest (RF) regression, an integrated modern machine learning regression algorithm based on classification trees, was proposed by Breiman [51] in 2001. The predicted results were averaged by integrating decision trees after the samples were constantly regressed and sampled several times to generate a training set. The algorithm must primarily optimize two essential parameters: ntree (number of decision trees) and mtry (number of segmentation nodes). In this study, ntree was set to 500, and mtry was set to 1/3 of the number of independent variables [52–54]. The RF regression was conducted based on the RF program package in R statistical software.

### 2.5.4. Accuracy Testing

The coefficient of determination ($R^2$), root mean square error (RMSE), and relative prediction deviation (RPD) were used to test the effects of the CCC estimation models. $R^2$ represents the degree of agreement between the predicted and measured values. The RMSE reflects the degree of deviation between the predicted and measured values. RPD is measured by the relative deviation between the predicted and measured values [55], which is the ratio between the standard deviation of the validation set and the RMSE [56]. The higher $R^2$, RPD, and lower RMSE indicate a better effect of the model [57]. It is believed that the model with RPD < 1.4 is unable to estimate the sample. When 1.4 < RPD < 2.0, it is considered that the model can roughly estimate the sample, and the predictive ability is acceptable. The model has an excellent predictive ability when RPD > 2.0 [58]. In this study, $R_c^2$, $RMSE_c$ and $R_v^2$, $RMSE_v$ represent $R^2$ and RMSE in the calibration and validation datasets, respectively.

## 3. Results

### 3.1. Sensitive-Band Reflectance (SR)-CCC Estimation Models

3.1.1. Correlations between Winter Wheat Canopy Reflectance and CCC

The canopy reflectance of winter wheat at the VIS band was more correlated with CCC than with NIR wavelengths, and the correlation coefficient between CSTs and CCC was higher than that of OS (Figure 2). There was a negative association between OS and CCC at 350–725 nm but a positive correlation at 726–1350 nm. The correlation coefficient between FDS and the CCC changed significantly at 350–1350 nm. It was moderately better at 555–635 nm than at other wavelengths. The CRS exhibited a substantial negative correlation with the CCC at 350–762 nm, while the correlation coefficient changed dramatically at 762–1350 nm. Based on the highest correlations, SRs at 683, 630, and 699 nm were selected as the sensitive bands for OS, FDS, and CRS, and the correlation coefficients were −0.81, −0.87, and −0.89, respectively.

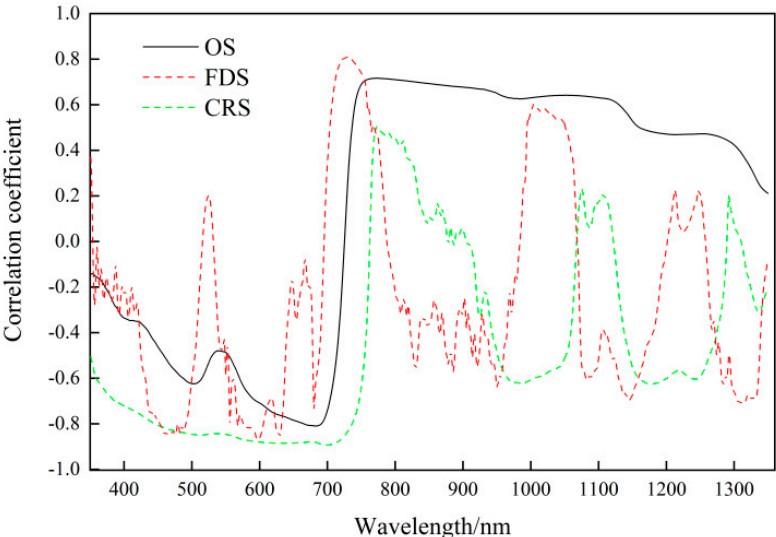

**Figure 2.** Correlation coefficient between CSTs (OS, FDS, and CRS) and CCC.

### 3.1.2. SR-CCC Estimation Models

SRs were used to predict the CCC (Figure 3) and the model validations were shown in Figure 4. The validation accuracy of FDS at 630 nm was similar to OS at 683 nm. The CRS at 699 nm enhanced the prediction accuracy of the CCC substantially compared to the OS model. The $R_c^2$ in the unary quadratic prediction models was 0.80 and the $R_v^2$ was 0.83. It evaluated the highest estimation accuracy with an RPD of 2.42. The scatter-distribution between the predicted and measured the CCC was closer to the 1:1 line (Figure 4).

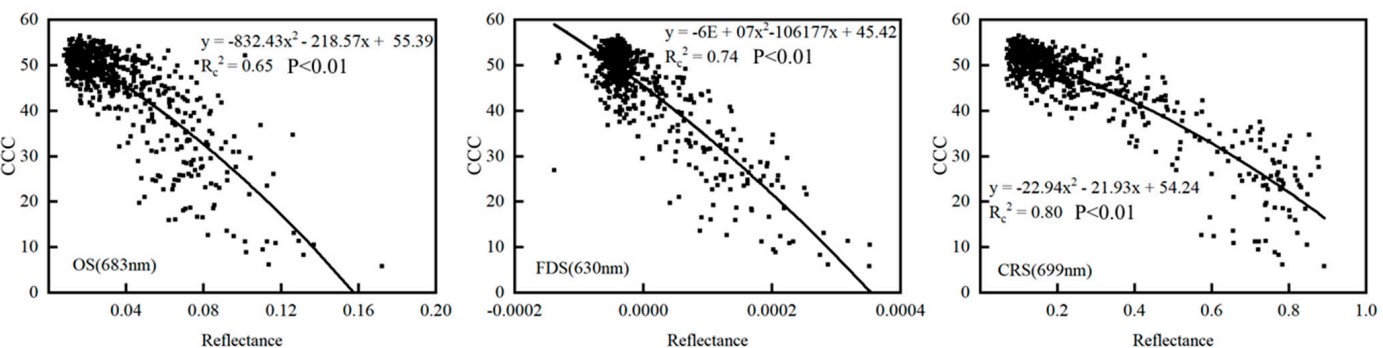

**Figure 3.** Fitting curve of SR-CCC estimation models.

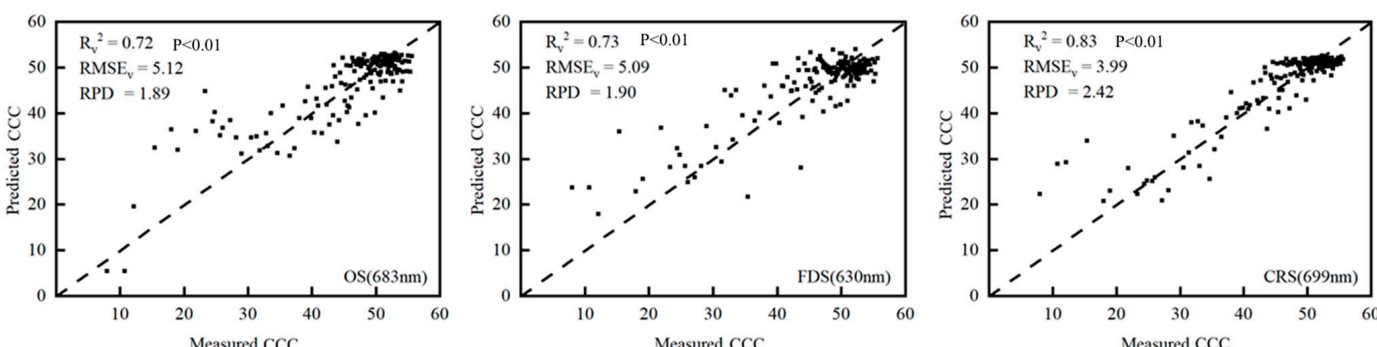

**Figure 4.** Measured and predicted CCC of SR-CCC estimation models.

### 3.2. Narrow-Band Spectral Indices (NSI)-CCC Estimation Models

#### 3.2.1. Correlation Analysis between NSI and CCC

The correlation equipotential maps between the NSI and CCC were constructed (Figure 5). The optimal band combinations of 12 spectral indices were selected, as shown in Table 4, when the correlation coefficient was the largest. All the correlation coefficients between NSIs and the CCC passed the significance test at the 0.01 level. The results showed that the optimal band combinations were principally located at 550–750 nm, and the absolute values of correlation coefficients were between 0.86 and 0.92. The correlation between the NSI and the CCC was significantly higher than that between the SR.

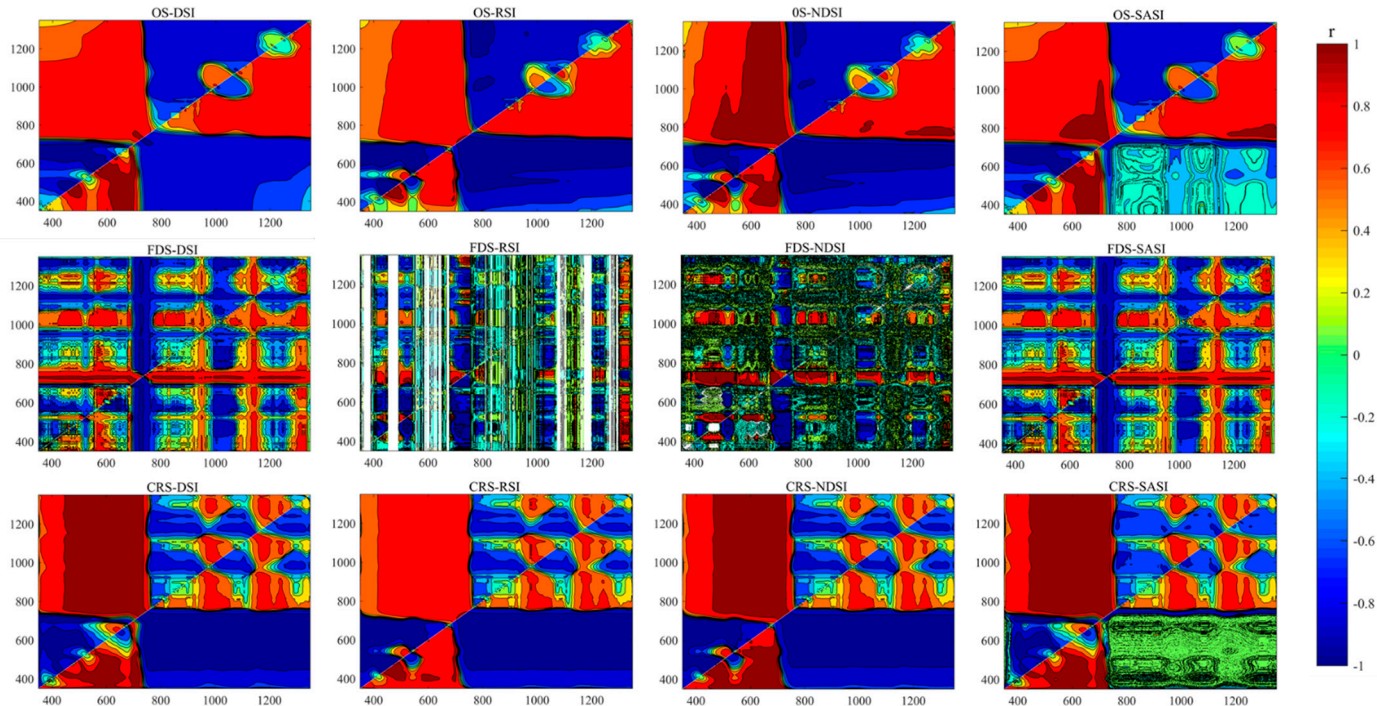

**Figure 5.** Equipotential diagram of the correlation coefficient between NSI and the CCC.

**Table 4.** Optimal band combinations of NSI for estimating CCC.

| Spectral Transformation | Spectral Indices | Correlation Coefficient |
|---|---|---|
| OS | DSI ($R_{595}$, $R_{695}$) | −0.91 ** |
| | RSI ($R_{564}$, $R_{701}$) | −0.87 ** |
| | NDSI ($R_{565}$, $R_{700}$) | −0.88 ** |
| | SASI ($R_{596}$, $R_{695}$) | −0.91 ** |
| FDS | DSI ($R_{609}$, $R_{639}$) | −0.86 ** |
| | RSI ($R_{741}$, $R_{1314}$) | 0.91 ** |
| | NDSI ($R_{686}$, $R_{724}$) | 0.91 ** |
| | SASI ($R_{609}$, $R_{639}$) | −0.86 ** |
| CRS | DSI ($R_{351}$, $R_{706}$) | −0.91 ** |
| | RSI ($R_{564}$, $R_{702}$) | −0.87 ** |
| | NDSI ($R_{366}$, $R_{698}$) | −0.89 ** |
| | SASI ($R_{353}$, $R_{720}$) | 0.92 ** |

Note: ** represents significance level *p* < 0.01.

#### 3.2.2. NSI-CCC Estimation Models

Here, the CCC was determined by the NSIs of the CSTs. First, based on OS, the calibration and validation of OS-DSI ($R_{595}$, $R_{695}$) had the same accuracy as OS-SASI ($R_{596}$, $R_{695}$), with slightly greater prediction accuracy than OS-RSI ($R_{564}$, $R_{701}$) and OS-NDSI ($R_{565}$, $R_{700}$). The $R_c{}^2$s in the unary quadratic prediction models were 0.82. In the validation set,

the $R_v^2$s were 0.84. RMSE$_v$s were 4.00 and RPDs were 2.41 for OS-DSI ($R_{595}$, $R_{695}$) and OS-SASI ($R_{596}$, $R_{695}$). OS-DSI ($R_{595}$, $R_{695}$) and OS-SASI ($R_{596}$, $R_{695}$) were better than the other OS-NSIs in predicting CCC, and the predicted and measured CCC values were close to the 1:1 line (Figures 6 and 7).

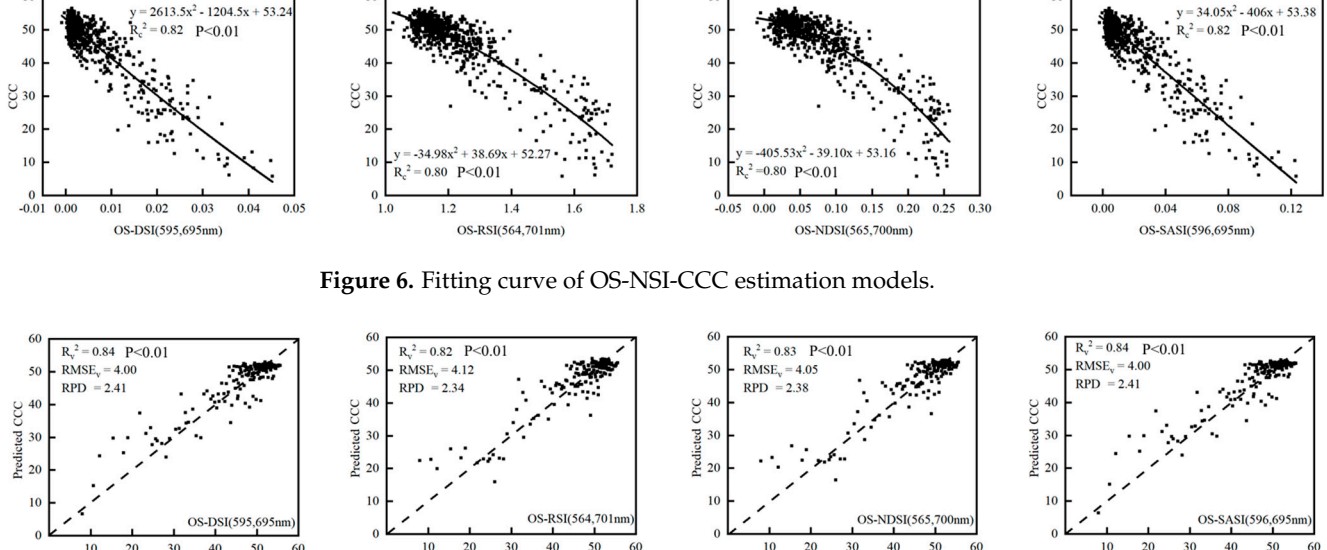

**Figure 6.** Fitting curve of OS-NSI-CCC estimation models.

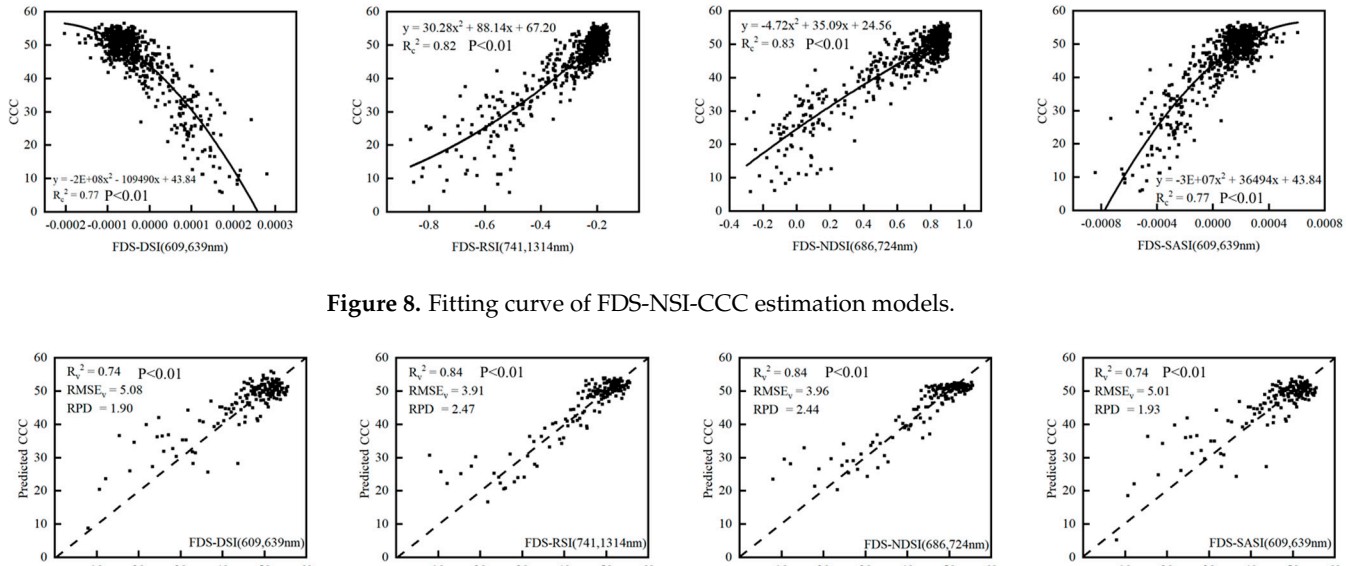

**Figure 7.** Measured and predicted CCC of OS-NSI-CCC estimation models.

Subsequently, as shown in Figures 8 and 9, the calibration and validation accuracies of FDS-DSI ($R_{609}$, $R_{639}$) were comparable to that of FDS-SASI ($R_{609}$, $R_{639}$). FDS-RSI ($R_{741}$, $R_{1314}$) and NDSI ($R_{686}$, $R_{724}$) significantly enhanced the prediction accuracy of the CCC compared to the FDS-DSI ($R_{609}$, $R_{639}$) and FDS-SASI ($R_{609}$, $R_{639}$) models. The $R_c^2$s in the unary quadratic prediction models were 0.82 and 0.83, respectively (Figure 8). The $R_v^2$s in the validation samples were 0.84 and 0.84. The RMSE$_v$s were 3.91 and 3.96; the RPDs were 2.47 and 2.44 for FDS-RSI ($R_{741}$, $R_{1314}$) and NDSI ($R_{686}$, $R_{724}$), respectively. FDS-RSI ($R_{741}$, $R_{1314}$) was superior to the other models in predicting CCC based on FDS-NSI, with predicted and measured CCC closer to the 1:1 line.

**Figure 8.** Fitting curve of FDS-NSI-CCC estimation models.

**Figure 9.** Measured and predicted CCC of FDS-NSI-CCC estimation models.

Finally, using the CRS-NDSI ($R_{366}$, $R_{698}$) to predict the CCC could account for 84% of the variability in the CCC (Figure 10). The high $R_v^2$ (0.87), RPD (2.73), and low $RMSE_v$ (3.53) also showed better performance of CRS-NDSI ($R_{366}$, $R_{698}$). This model was superior to other models in predicting CCC-based CRS-NSI and the predicted and measured CCC values were close to the 1:1 line (Figure 11).

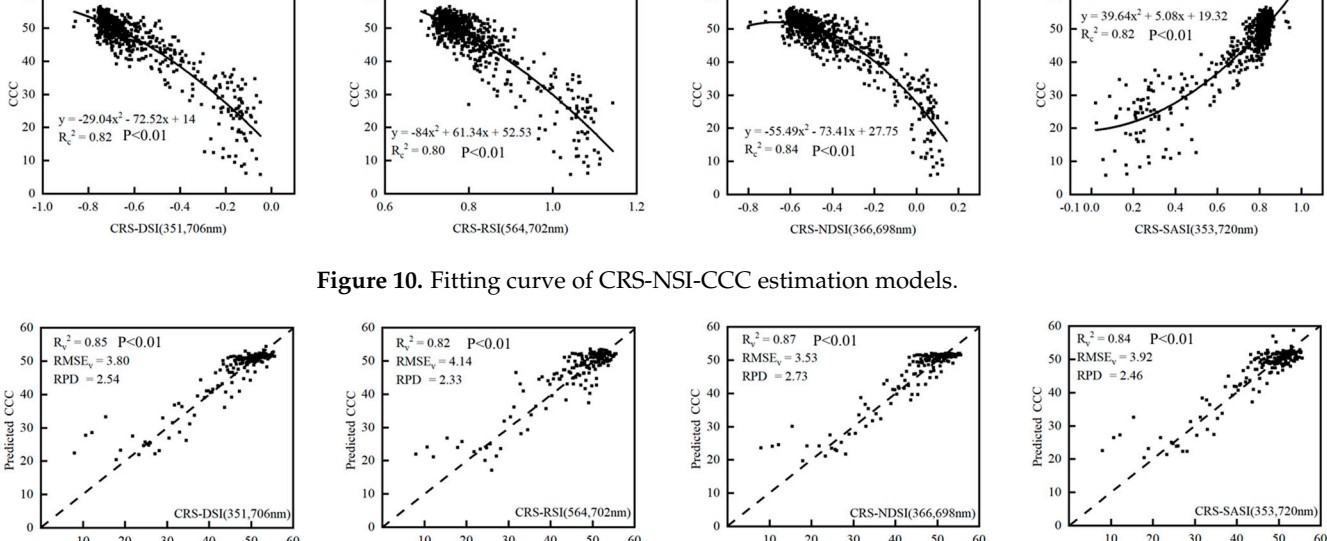

**Figure 10.** Fitting curve of CRS-NSI-CCC estimation models.

**Figure 11.** Measured and predicted CCC of CRS-NSI-CCC estimation models.

### 3.3. PLS-CCC and RF-CCC Estimation Models

CCC estimation models were established using multivariate analysis based on the PLS and RF regression. In this study, all SRs and NSIs based on CSTs were regarded as multivariate parameters. All the models had $R_c^2$s larger than 0.80 (Table 5). The prediction accuracy of the FDS-RF-CCC model was the same as that of the CRS-RF-CCC model, and the $R_c^2$ was 0.97. However, the FDS-RF-CCC in the validation set had superior $R_v^2$ (0.88) and RPD (2.88), as well as a lower $RMSE_v$ (3.35). The FDS-RF-CCC performed the best of all models with the predicted and measured CCC closer to the 1:1 line (Figure 12).

### 3.4. Model Precision Comparison

Compared to OS-SR (683 nm) and FDS-SR (630 nm), CRS-SR (699 nm) significantly enhanced CCC estimation accuracy. The $R_c^2$, $R_v^2$, and RPD increased to 0.80, 0.83, and 2.42, respectively, and the RMSEv was 3.99 (Figures 3 and 4). Of all the univariate models, the CRS-NDSI-CCC model had the best performance ($R_c^2$ = 0.84, $R_v^2$ = 0.87, $RMSE_v$ = 3.53, and RPD = 2.73).

Among the multivariate spectrum parameter regression models, FDS stood out from the CSTs and had good potential for predicting the CCC. In the RF regression, the FDS-RF-CCC model yielded the highest prediction accuracy in both the calibration and validation sets ($R_c^2$ = 0.97, $R_v^2$ = 0.88, $RMSE_v$ = 3.35, and RPD = 2.88). In the PLS regression, the FDS-PLS-CCC model showed the best performance in predicting the CCC, with $R_c^2$, $R_v^2$, $RMSE_v$, and RPD being 0.86, 0.88, 3.43, and 2.82, respectively, which were slightly poorer than the FDS-RF-CCC model.

The RPD values of all CCC estimating models were shown in Figure 13. Except for the OS-SR-CCC, FDS-SR-CCC, FDS-DSI-CCC, and FDS-SASI-CCC models, the RPDs of all the other estimation models were more than 2.0, indicating that the majority of the estimation models had accurate predictive abilities. Overall, when all of the models in this study were compared using statistical indicators such as $R^2$, RMSE, and RPD, an integrated strategy based on FDS and RF regression demonstrated the highest accuracy in CCC estimations, followed by the FDS-PLS-CCC model.

**Table 5.** $R_c^2$ of PLS-CCC and RF-CCC models.

| Models | OS-PLS-CCC | FDS-PLS-CCC | CRS-PLS-CCC | OS-RF-CCC | FDS-RF-CCC | CRS-RF-CCC |
|---|---|---|---|---|---|---|
| $R_c2$ | 0.84 | 0.86 | 0.85 | 0.96 | 0.97 | 0.97 |

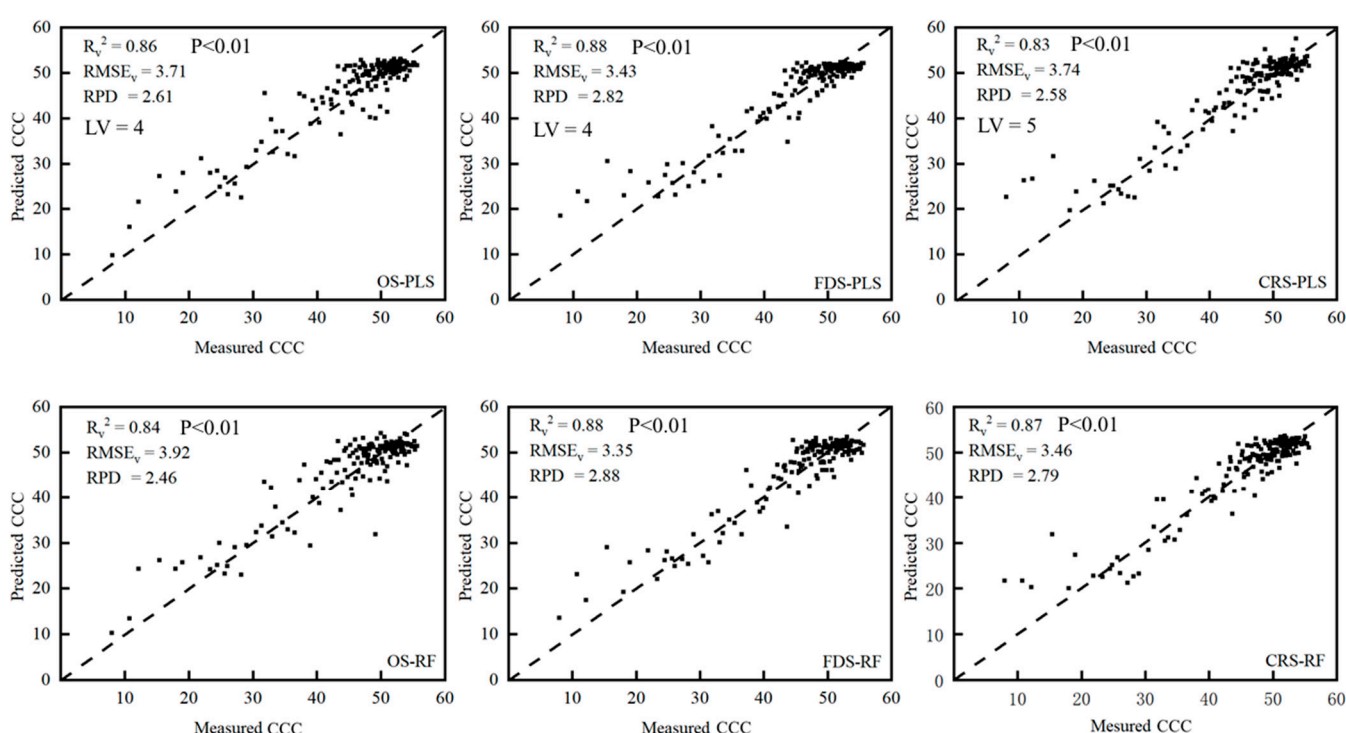

**Figure 12.** Measured and predicted CCC of PLS and RF regression models.

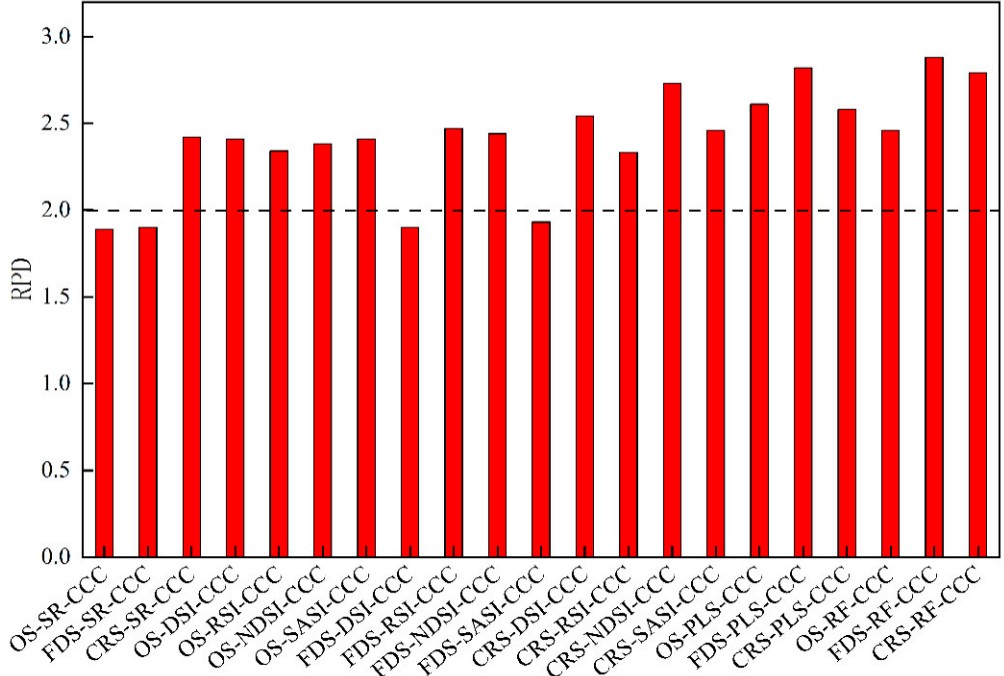

**Figure 13.** Relative prediction deviation (RPD) of CCC prediction models.

## 4. Discussion

Chlorophyll in plants absorbs light energy and is involved in photosynthesis to produce chemical energy. Effective monitoring of chlorophyll content is of great importance to control the fertilization of plants, detect the health status of plants, and estimate yield. Many studies have been conducted on the rapid and accurate monitoring of canopy chlorophyll content (CCC) using remote sensing [59–61]. This study primarily focused on the influence of VIS and NIR reflectance (350–1350 nm) on estimating CCC, extracting SR and NSI by CST to construct the CCC estimation models to provide scientific methods and technical guidance for planting winter wheat in the Guanzhong Plain area.

### 4.1. Extraction of SR and NSI

In this study, a hyperspectral canopy ground dataset in situ was measured to estimate the CCC of winter wheat and yielded significant results. To the best of our knowledge, due to the effect of canopy chlorophyll, the reflectance of crops is strongly absorbed near the red band and forms an absorption valley, while the strong reflection near the near-infrared band reaches a high reflection peak. The red-edge position (REP) is located at the position where the curve slope changed the most during the transition from the red band to the near red band [62]. Based on OS, FDS, and CRS, we determined the NSI (DSI, RSI, NDSI, and SASI) to select the most suitable for estimating CCC. As shown in Table 4, almost all bands used to construct NSI contained a band near REP (680–760 nm), indicating the REP had a significant response to CCC. This may be because REP was located in the middle of the strong absorption band of the red-edge band of the vegetation and the strong reflection area of the NIR, which effectively expanded the response of green vegetation to CCC. Previous studies on chlorophyll correlation have also highlighted the importance of the red-edge bands [63,64], and our results were in agreement. In addition, REP is also very important in estimating various physical and chemical parameters of crop plants [65–69]. Li et al. [70] showed the mSR ($mSR_{705}$) index constructed using the red-edge band had the best performance among all spectral indices for evaluating nitrogen concentration in winter wheat. Su et al. [71] found the normalized spectral index ($NDSI_{705,783}$) based on the Sentinel-2 remote sensing image, including the red-edge band, could predict the leaf area index of maize more accurately. Therefore, it can be predicted that the introduction of the red-edge band in the estimation and inversion of physical and chemical parameters of crops will improve the efficiency and capability of these methods in the future.

### 4.2. Canopy Spectral Transformation (CST)

Previous studies had revealed more sensitive features could be provided and the accuracy of estimating the physical and chemical parameters of crop plants could be greatly improved by transforming OS [41,72]. Ren et al. [73] improved the response characteristics of the winter wheat canopy spectrum to low-temperature stress by FDS, which is because FDS can effectively eliminate the background noise of soil and consider overlapping spectral features [58]. In estimating the nitrogen concentration of leaves, CRS could maximize the prediction accuracy [74] and effectively extend the difference in absorption intensity [41]. In this study, SR and NSI were extracted based on the OS, FDS, and CRS. As shown in Figures 3 and 4, CRS achieved the best performance in the CCC estimation model based on the SR. As demonstrated in Figures 6–11, in CCC estimation models based on NSIs, the NDSI-CCC model extracted based on CRS had the best accuracy. This was better than FDS and OS, and was consistent with the results of Ramoelo et al. [75]. Among the CCC prediction models, FDS-RF-CCC exhibited the best accuracy in CCC estimation, which was constructed based on the FDS. Overall, the results showed CST could provide more sensitive features than OS, and it was possible by transforming OS to estimate the physical and chemical parameters of the plants.

*4.3. Canopy Chlorophyll Content (CCC) Estimation Models*

The modeling results of this study showed NSI could generally improve the accuracy of the CCC estimation model (Figures 6–11). The best hyperspectral NSI (NDSI ($R_{366}$, $R_{698}$)) based on CRS showed a 4% higher variability than the best SR (CRS (699 nm)). Multivariate regression had significant advantages compared to the CCC univariate estimation model (Table 5). The multivariate estimation models PLS-CCC and RF-CCC based on FDS had the best prediction accuracy (Table 5). However, by analyzing the results of the accuracy of the validation models, it was found the PLS-CCC model was slightly worse than the RF-CCC model, which was consistent with the conclusion of Wang et al. [76] in estimating the leaf area index of rice. The main reason was that the PLS regression model was not as robust as the RF regression model. As one of the widely used machine learning methods, the RF regression model can better cope with disturbances and outliers owing to its internal mechanism of majority voting. When using the RF regression model, overfitting did not occur easily as long as we adjusted the key parameters precisely, which was more suitable for solving some nonlinear problems than the PLS regression model [45].

Most of the RPD values in Figure 13 were greater than 2.0, indicating the CCC prediction models based on different multivariate regression methods were excellent. In this study, it was found that FDS and RF regression together overcame the limitations of the univariable regression technique and provided a practical method for monitoring the CCC of winter wheat by comparing the prediction accuracy of all models. The FDS-RF-CCC had the best performance ($R_v^2$ = 0.88, $RMSE_v$ = 3.35, and RPD = 2.88) and was a promising method. Although the CRS-NDSI-CCC model (RPD = 2.73) can also be used to predict CCC when the prediction accuracy requirement is not particularly accurate, it is still necessary to use the new prediction model (FDS-RF-CCC) proposed in this paper to better meet the needs of precision agriculture in the Guanzhong Plain area.

*4.4. Future Works*

In this study, the CST was used in combination with univariate, PLS, and RF regression methods to analyze the performance of the winter wheat CCC models. The combination of FDS and RF regression explained the wavelengths that were helpful in creating the model with the best predictive accuracy for the winter wheat CCC. The FDS-RF-CCC model proposed in this study can be used to quantitatively estimate winter wheat CCC in the Guanzhong Plain area. Most previous studies have shown a strong link between crop chlorophyll content and nitrogen content, having used this relationship to estimate crop nitrogen [67,77–79]. Homolová et al. [5] strongly supported the hypothesis that optical remote sensing of chlorophyll content could be used as a substitute for nitrogen estimation based on a moderate to good relationship between nitrogen and chlorophyll. Schlemmer et al. [67] pointed out that the chlorophyll content establishes a certain relationship between remote sensing observations and canopy state variables, and this relationship was used to indicate the nitrogen concentration of corn and the ability of photosynthesis. It was also confirmed there was a very close relationship between chlorophyll content and nitrogen concentration [6]. As a result of the quantitative estimation of winter wheat CCC, this study provides a technical method for field crop nitrogen assessment, as well as a theoretical basis for precise agricultural nitrogen management in the future.

However, the FDS-RF-CCC model was constructed based on specific winter wheat varieties in the Guanzhong Plain area. Whether this model can be successfully applied and is suitable for estimating CCC in different varieties needs to be further investigated.

**5. Conclusions**

Canopy chlorophyll content (CCC) is closely related to crop nitrogen status crop growth, detection of diseases and pests, and final yield. Thus, accurate monitoring of chlorophyll content in crops is of great significance for decision support in precision agriculture. In this study, winter wheat in the Guanzhong Plain area of Shaanxi Province, China, was selected as the research subject to explore the feasibility of canopy spectral transfor-

mation (CST) combined with a machine learning method to estimate CCC. The results showed the reliability of CST combined with machine learning method to estimate winter wheat CCC. Compared with OS-SR (683 nm), FDS-SR (630 nm) and CRS-SR (699 nm) had a larger correlation coefficient between canopy reflectance and CCC. Among the parametric regression methods, the univariate regression method with CRS-NDSI as the independent variable achieved satisfactory results in estimating the CCC of winter wheat. As a machine learning method, RF regression combined with multiple independent variables showed the best winter wheat CCC estimation accuracy. The FDS-RF-CCC had the best accuracy in estimating CCC, providing a promising method for rapid and nondestructive estimation of winter wheat CCC. Future research should focus on further optimization of the FDS-RF-CCC model and apply this model to the precision nitrogen management of winter wheat.

**Author Contributions:** Conceptualization, X.C.; methodology, X.C.; software, X.C.; validation, X.C., Q.C., and F.L.; formal analysis, X.C. and B.S.; investigation, X.C., K.F. and Z.L.; resources, Q.C.; data curation, X.C. and B.S.; writing—original draft preparation, X.C.; writing—review and editing, X.C.; visualization, X.C.; supervision, Q.C. and F.L.; project administration, F.L.; funding acquisition, Q.C. All authors have read and agreed to the published version of the manuscript.

**Funding:** This research was funded by the National Natural Science Foundation of China (41701398).

**Data Availability Statement:** Data sharing is not application to this article.

**Conflicts of Interest:** The authors declare no conflict of interest.

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
