# Peer review of "Estimation of Winter Wheat Canopy Chlorophyll Content Based on Canopy Spectral Transformation and Machine Learning Method"

_agronomy, doi:10.3390/agronomy13030783_

Round 1

Reviewer 1 Report

Overall, the paper is of good quality, sufficient for publishing in the journal without revision. It should be noted that it goes deep into technical peculiarities of the subject.
The main concern expressed by the authors: "Whether this model can be successfully applied and is suitable for estimating CCC in different varieties and different ecological locations needs to be further investigated." This is indeed an important point which can be extended further with questions like: "How does the proposed model work in the case of non-typical weather conditions including seasonal anomalies?" or "Can it take into account global climate changes likes warming effects?" But, as the authors state, such questions are beyond the scope of the paper.

Reviewer 2 Report

The estimation of CCC is critical for wheat crop management. The current work is interesting and practical. However, this article has many spelling and formatting errors, too many old references, improper chart editing, and very limited innovation. Hence, it needs to have the related works reviewed, the innovations added, and the wording and expression polished. I cannot recommend it for further review in its current version. major concerns: -1, L89-91. Why does it "is worth discussing"? Didn't previous research focus on this discussion? Please refer to the following articles for discussion. [1] X. Zhang, H. Sun, X. Qiao, X. Yan, M. Feng, L. Xiao, X. Song, M. Zhang, F. Shafiq, W. Yang, C. Wang, Hyperspectral estimation of canopy chlorophyll of winter wheat by using the optimized vegetation indices, Computers and Electronics in Agriculture, 193 (2022). -2, L101. What is the modern machine learning method in this study? According to a study in PNAS (Belkin et al., 2019), "However, practitioners routinely use modern machine-learning methods, such as large neural networks and other nonlinear predictors that have very low or zero training risk." Obviously, the univariate regression and PLS regression are linear models. Please clarify. [1] M. Belkin, D. Hsu, S. Ma, S. Mandal, Reconciling modern machine-learning practice and the classical bias-variance trade-off, Proc Natl Acad Sci U S A, 116 (2019) 15849-15854. -3, Please number all subgraphs in Figures 3-12. Please conduct significance test and add significance level (p value) for all modeling and validation. Please add the optimal number of latent variables for each PLS regression in Figure 12. -4, In this study, the principal research techniques used are spectral filtering, the first-order derivative, envelope removal, correlation analysis, the two-band index, univariate regression, partial least squares regression, and random forest regression. However, these spectral processing technologies are very common and hardly new. So, it is doubtful to call the combination of these technologies innovation.   -5, Of the 78 references, only one is in 2022, and four are in 2021, most of which are before 2020 or even before 2010. However, a simple search (keywords: "CCC; wheat; spectra" in web of science) can see several cutting-edge works that are similar to the current work in this paper, for examples: [1] X. Zhang, H. Sun, X. Qiao, X. Yan, M. Feng, L. Xiao, X. Song, M. Zhang, F. Shafiq, W. Yang, C. Wang, Hyperspectral estimation of canopy chlorophyll of winter wheat by using the optimized vegetation indices, Computers and Electronics in Agriculture, 193 (2022). [2] Q. Jiao, Q. Sun, B. Zhang, W. Huang, H. Ye, Z. Zhang, X. Zhang, B. Qian, A Random Forest Algorithm for Retrieving Canopy Chlorophyll Content of Wheat and Soybean Trained with PROSAIL Simulations Using Adjusted Average Leaf Angle, Remote Sensing, 14 (2021). I seriously doubt that the authors had done an in-depth job of tracking and reviewing the related articles.   minor concerns: -1, L16. What exactly does "over 3 years from 2014 to 2017" mean? 3 growth seasons? -2, L62-63,65. wrong acronym "VIS" -3, L88: "RF"; L93, "PLS" . Please ensure that the first written abbreviation in the abstract and the main text includes the full name. -4, L192. formatting error -5, L214. "innovativeregression"?

Round 2

Reviewer 2 Report

The authors have given proof that this article has been reviewed by native speakers. I have chosen to believe it. I have no further comments for the authors; they have replied to all that I asked. Thank you again.